# Convolutional neural networks for mesh-based parcellation of the cerebral cortex

**Guillem Cucurull**[1,2], **Konrad Wagstyl**[3,4], **Arantxa Casanova**[1,2], **Petar Veličković**[1,5],
**Estrid Jakobsen**[4], **Michal Drozdzal**[1,6], **Adriana Romero**[1,6], **Alan Evans**[4], **Yoshua Bengio**[1]

[1] Montreal Institute for Learning Algorithms, Montreal, QC, Canada
[2] Centre de Visió per Computador, UAB, Spain
[3] Department of Psychiatry, University of Cambridge, UK
[4] Montreal Neurological Institute, Montreal, QC, Canada
[5] Department of Computer Science and Technology, University of Cambridge, UK
[6] Facebook AI Research

## Abstract

In order to understand the organization of the cerebral cortex, it is necessary to create a map or parcellation of cortical areas. Reconstructions of the cortical surface created from structural MRI scans, are frequently used in neuroimaging as a common coordinate space for representing multimodal neuroimaging data. These meshes are used to investigate healthy brain organization as well as abnormalities in neurological and psychiatric conditions. We frame cerebral cortex parcellation as a mesh segmentation task, and address it by taking advantage of recent advances in generalizing convolutions to the graph domain. In particular, we propose to assess graph convolutional networks and graph attention networks, which, in contrast to previous mesh parcellation models, exploit the underlying structure of the data to make predictions. We show experimentally on the Human Connectome Project dataset that the proposed graph convolutional models outperform current state-of-the-art and baselines, highlighting the potential and applicability of these methods to tackle neuroimaging challenges, paving the road towards a better characterization of brain diseases.

## 1 Introduction

The cerbral cortex is the large multilayered, folded structure on the outer surface of the human brain. Different areas of the cortex are involved in many of our complex cognitive processes, including high-order visual processing, language and social interactions. Damage to the cortex can therefore lead to a wide variety of neurological and neuropsychiatric conditions. One longstanding challenge in understanding how the cerebral cortex is organized is to create a map of these areas, and parcellate it into functionally and structurally discrete areas.

Historically, parcellation has been based on expert examination of 2D cortical sections. Areal distinctions were made based on the cytoarchitecture (based on patterns of neurons) [4], myeloarchitecture (based on the distribution of wiring-related myelin) [38] and various other post mortem measures of microstructural organization [28]. However, a map of cortical areas is best comprehended on a surface, and there are numerous difficulties in creating a cortical surface parcellations based on borders identified on a limited number of 2D post mortem sections.

With the advent of 3D neuroimaging, it has been possible to create in vivo mesh reconstructions of individuals' cortical surfaces. A large part of neuroimaging is now carried out on such meshes [22, 10]. Cortical meshes create a common coordinate system to represent multiple modalities,

Submitted to 1st Conference on Medical Imaging with Deep Learning (MIDL 2018), Amsterdam, The Netherlands.

including structural and functional Magnetic Resonance Imaging (MRI), diffusion MRI and Positron Emission Tomography (PET). The same mesh structure can be used to represent a variety of different features, like cortical thickness, curvature and functional connectivity extracted from imaging volumes. Moreover, morphological and functional similarities can be used to coregister cortical surfaces between individuals. Thus, meshes are commonly used to investigate the structural, functional and developmental patterns of a healthy brain, alongside abnormalities in neurological and psychiatric conditions. Therefore, mesh-based analyses of cortical data capture the interindividual and interareal variance of the cortex that is of scientific interest. These interareal differences in multimodal signals, mapped to the cortical surface, can thus be used to parcellate the cortex.

There have been numerous attempts to segment the cortical mesh using multimodal in vivo data. Jakobsen et al. [21] proposed to compare functional connectivity maps to group-average patterns of area-of-interest connectivity. Glasser et al. [12] approached the mesh segmentation problem by training a multi-layer perceptron on node features. However, these methods consider each vertex of the mesh independently and do not exploit the underlying graph structure of the data.

At the same time, in the last decade, we have experienced remarkable advances in the fields of computer vision, medical imaging, speech recognition and natural language processing. These advances have mainly been driven by the successful design and application of deep learning architectures. In medical imaging, Convolutional Neural Networks (CNNs) have been adapted and extended to tackle relevant research challenges, including the segmentation of biomedical images, where the underlying data representation has a grid-like structure. However, many researchers deal with irregular data that could be represented using graph or mesh structures and the application of CNNs to such data poses different challenges.

Early approaches to leverage neural networks for processing graph structured data used recursive neural networks on directed acyclic graphs [11, 34]. These approaches were further generalized and improved by Graph Neural Networks [33, 14, 25] to deal with a more general class of graphs. Moreover, methods relying on the node features complemented with node similarity constraints [39] or structural features [29, 40] were also introduced in the literature. In recent years, we have experienced increasing interest in generalizing convolutions to the graph domain. Efforts in this directions include spectral approaches such as [5, 17], which work with a spectral representation of the graphs and define the convolution operation in the Fourier domain. These approaches were improved in [6, 24] by mitigating the computations and making the convolutional filters localized. Moreover, efforts have also been devoted to develop non-spectral approaches such as [8, 3, 15], which define convolutions directly on the graph and operate on groups of spatially close neighbors. One of the challenges of these approaches is to define an operator which works with different sized neighborhoods and maintains the weight sharing property of CNNs. This problem was recently addressed in [37] by means of an attention-based architecture reminiscent of [18, 32, 26], which yields top performance across several benchmarks.

In this paper, we study cortical meshes parcellation into brain areas using the Human Connectome Project data. More specifically, we apply recent deep learning models to segment two adjacent cytoarchitectonic areas, 44 and 45, inside Broca's area [20]. Studying Broca's area is interesting because it has an important role in language processing [31]. Traditionally, in neuropsychological research, morphologically defined regions of interest have been used as proxies for areas 44 and 45 [21], but the high degree of variability between different subjects requires models that produce subject-specific segmentations if a more precise localization of each area is needed.

We approach cortical meshes parcellation as a graph segmentation problem, where the model receives a mesh as input and produces one output label for each node of the mesh. To process cortical meshes, we adapt the recently proposed Graph Convolutional Networks [24] and Graph Attention Networks [37] and compare them to several simple baselines, which are agnostic to the graph structure. We evaluate all the methods in terms of Jaccard score as well as visual inspection. The results show that the methods that operate directly in the graph domain are able to exploit the underlying structure of the data improving the parcellation performance when compared to alternative approaches and baselines. Moreover, we report state-of-the-art results on the task of Broca's area parcellation, when using either Graph Convolutional Networks or Graph Attention Networks.

# 2 Models

In this section, we describe different variants of the models that are suitable to tackle the problem of cortical mesh segmentation. First, we discuss simple baseline approaches. Then, we review two recent graph convolution methods: Graph Convolutional Networks and Graph Attention Networks.

## 2.1 Baselines

**NodeMLP** [2]: The NodeMLP baseline frames the mesh segmentation as node classification problem, where we classify each node in the mesh separately. The model consists of a shared multi-layer perceptron that processes each node in the cortical mesh independently, and produces a label prediction for each one of them. Hence, the output prediction for each node *depends on its features exclusively*, completely disregarding positional or neighborhood information.

**NodeAVG:** The NodeAVG baseline ignores the node features and *relies on the position of each node exclusively* to provide a segmentation output. The baseline computes each node's most frequent label, based on the meshes of the training set, and predicts the associated most frequent label at test time. Note that the applicability of this model is *limited* to tasks where all the meshes have the same topology. By comparison with the NodeMLP [2] baseline, NodeAVG allows us to assess how critical the position information of a node is; note that, in this case, the position is the only feature.

**MeshMLP:** The MeshMLP baseline *jointly processes all the nodes* in a mesh—and produces as output a segmentation prediction for each one of them. This model processes the whole mesh at once, by flattening it as a vector, which discards all structural information and concatenates the features of each node. The flattened mesh representations are fed to a multi-layer perceptron, which jointly processes all the nodes and produces as output a segmentation prediction for all the nodes in the mesh. Although structural information is discarded, this model is capable of exploiting global mesh information, which can influence the output of each node.

## 2.2 Graph Convolutional Networks

Graph Convolutional Networks (GCNs) [6, 24] are specifically designed to operate on graphs, thus, explicitly exploiting the underlying graph structure of the data. To do so, GCNs consider spectral convolutions on graphs defined as the multiplication of a signal with a filter in the Fourier domain [5, 17]. It follows that signal $h_i$ of node $i$, linearly transformed by $\mathbf{W}$, is filtered by $g$ as:

$$g \star \mathbf{W} h_i = \mathbf{U}(\mathbf{U^T} g \odot \mathbf{U^T} \mathbf{W} h_i), \tag{1}$$

where $\mathbf{U}$ is the Fourier basis of the graph Laplacian $\mathbf{L}$ (obtained by computing the eigendecomposition of the latter), $\star$ is the convolution operator and $\odot$ is the elemen-wise multiplication.

To yield spatially localized filters and remove the need to compute the eigendecomposition of $\mathbf{L}$, GCNs approximate the filters by means of a truncated expansion of Chebyshev polynomials of the graph Laplacian up to order $K$. Therefore, each GCN layer is graph convolutional layer that takes as input a graph and produces a graph as output. Given a feature vector $h_i$ of node $i$, the output of a graph convolutional layer $h'_i$ is computed as follows:

$$h'_i = \sum_k^K w_k T_k(\mathbf{L}) \mathbf{W} h_i, \tag{2}$$

where $w_k$ are the Chebyshev coefficients, $T_k$ the Chebyshev polynomial of order $k$ and $\mathbf{W}$ the parameters of a learnable transformation applied to $h_i$. Thus, the output of a graph convolutional layer depends only on information from a local neighborhood around it (up to $k$ steps away from the central node). This local neighborhood can be increased by stacking several layers on top of each other, allowing to exploit contextual information, and making a node's segmentation output depend on a larger part of the graph.

## 2.3 Graph Attention Networks

We also consider the recent Graph Attention Network (GAT) model [37], wherein the propagation layers have identical input-output structure as the GCN, but they specify the convolution weights

*implicitly* rather than explicitly. This property is achieved by leveraging a content-based self-attentional mechanism [36] which is restricted to attending only along the edges of the provided graph. As a consequence, the layer no longer depends on knowing the graph Laplacian upfront—it becomes capable of handling *inductive* as well as transductive graph prediction problems. Furthermore, the implicit specification of weights allows for trivially assigning different (learnable) importances to different nodes in a $k$-hop neighborhood, which is not possible with techniques such as the GCN.

We leverage the same self-attention mechanism as used in [37]. We compute the convolutional weights by applying a shared attentional mechanism $a : \mathbb{R}^F \times \mathbb{R}^F \to \mathbb{R}$ which computes *attention coefficients*

$$e_{ij} = a(h_i, h_j) \tag{3}$$

that indicate the *importance* of node $j$'s features to node $i$. We only compute $e_{ij}$ for nodes $j \in \mathcal{N}_i$, where $\mathcal{N}_i$ is some *neighborhood* of node $i$ in the graph. To make coefficients easily comparable across different nodes, we normalize them across all choices of $j$ using the softmax function:

$$\alpha_{ij} = \text{softmax}_j(e_{ij}) = \frac{\exp(e_{ij})}{\sum_{k \in \mathcal{N}_i} \exp(e_{ik})}. \tag{4}$$

The attention mechanism $a$ is a single-layer feedforward neural network, parametrized by a weight vector $\mathbf{a} \in \mathbb{R}^{2F'}$, and applying the LeakyReLU nonlinearity (with negative input slope $\alpha = 0.2$). Fully expanded out, the coefficients computed by the attention mechanism may then be expressed as:

$$\alpha_{ij} = \frac{\exp\left(\text{LeakyReLU}\left(\mathbf{a}^T[\mathbf{W}h_i \| \mathbf{W}h_j]\right)\right)}{\sum_{k \in \mathcal{N}_i} \exp\left(\text{LeakyReLU}\left(\mathbf{a}^T[\mathbf{W}h_i \| \mathbf{W}h_k]\right)\right)} \tag{5}$$

where $\cdot^T$ represents transposition, $\|$ is the concatenation operation, and $\mathbf{W}$ is a shared, learnable, transformation matrix.

Once obtained, the normalized attention coefficients are used to compute a linear combination of the features corresponding to them, to serve as the final output features for every node (after potentially applying a nonlinearity, $\sigma$):

$$h'_i = \sigma\left(\sum_{j \in \mathcal{N}_i} \alpha_{ij} \mathbf{W}h_j\right). \tag{6}$$

To stabilize the learning process of self-attention, we have found extending the mechanism to employ *multi-head attention* to be beneficial, similarly to [36]. Specifically, $K$ independent attention mechanisms execute the transformation of Equation 6, and then their features are concatenated.

## 3 Experiments

We evaluate the convolutional approaches against previous state-of-the-art and the proposed baselines on the Human Connectome Project dataset[1]. This section summarizes the dataset, our experimental setup and obtained results.

### 3.1 Dataset

The data used for the experiments comes from the Human Connectome Project (HCP) [13], consisting of 100 different subjects, with one mesh per subject. The nodes of the meshes have been manually annotated as in [20], assigning each node into one of the following labels: area 44, area 45 or neither. All the meshes from different subjects have the same structure, so they can be represented with the same adjacency matrix. Each mesh has 1195 nodes, representing Broca's area of the left hemisphere of the cerebral cortex. Each node of one mesh has 9 real valued features: 6 structural features (cortical thickness, myelin, curvature, sulcal depth, folding corrected cortical thickness and bias-corrected myelin) and 3 functional features (rsfMRI correlation with anterior temporal and two parietal regions of interest [20]), and each node has a single label, corresponding to the region of interest it belongs to.

---

[1]http://www.humanconnectomeproject.org/

Given the limited size of the dataset, we report results obtained by 10-fold cross validation of the models. We split the data such that eight folds are used for training, one for validation and the remaining one for test. We repeated this process 10 times (one per fold) and report the means and standard deviations of results on the different test sets. Results are reported in terms of per-class and average Jaccard index of the two classes of interest.

## 3.2 Experimental setup

The goal of our experimental section is to compare how different approaches perform in the task of cortical mesh segmentation, and to test if explicitly using the mesh structure of the data, with convolutional neural networks for graphs, improves the performance over methods that do not use that information. For all models, we have experimented with different configurations, selecting the best architecture and optimization hyper-parameters in terms of validation Jaccard index. Note that the Jaccard score is computed as the mean per-class Jaccard of the 2 classes of interest. Then, the best configuration results are reported for the test set.

All models are trained using backpropagation, with the Adam [23] optimizer. The optimized loss function is either the dice loss [7], extended to multiple classes by averaging class-specific dice losses, or a crossentropy loss, according to the best validation results. Moreover, given the class imbalance, when using cross-entropy loss, each node is weighted depending on their ground truth class. The weight assigned to each class $c$ is defined as $w_c = median\_freq(c)/freq(c)$, where $freq(c)$ is the number of nodes belonging to class $c$ divided by the total number of nodes, and $median\_freq(c)$ is the median of those frequencies. This class weighting is usually used in image semantic segmentation problems [9]. Finally, unless stated otherwise, the vector of features for each node is normalized so that it has unit norm.

**NodeMLP training details**: This model takes as input the 9 features of a node and stacks 4 fully connected layers on top of it, and a classification layer (with 3 possible outputs). Each hidden layer has 128 units, followed by a ReLU [27] non-linearity. Additionally, batch normalization [19] and dropout [35] with $p = 0.5$ are applied at the output of each hidden layer. The model is trained using a weighted cross-entropy loss, as described in the previous paragraph.

**MeshMLP training details**: This model takes as input a 10755-dimensional vector that concatenates the 9 features of all nodes in the mesh (1195 nodes), and stacks a hidden layer of size 32 and a classifier, producing as output a 3-dimensional vector per node. Hidden layers apply a ReLU non-linearity and are followed by both batch normalization and dropout with $p = 0.5$. The model is trained by minimizing a weighted cross-entropy loss.

**GCN training details**[2]: This model takes as input a mesh and outputs a label prediction for each node in the mesh. The architecture has 8 convolutional layers and uses a degree of $K = 8$ in the Chebyshev approximation described in Section 2.2. Each layer consists of 64 units followed by ReLU non-linearity. Additionally, we apply batch normalization after each layer. The model is trained with average dice loss across classes, which has proven to achieve better validation performance in this case.

**GAT training details**[3]: This model takes as input a mesh and outputs a label prediction per node. We apply an eight-layer GAT model. Each layer consists of $K = 8$ attention heads computing $F = 32$ features (for a total of 256 features). Unlike [37], each layer is followed by a batch normalization and a ReLU non-linearity and we increase the neighborhood masks to compute attention coefficients for all neighbors within 5-hops of the central node. Furthermore, dropout with p = 0.1 is applied to both layers' inputs, as well as to the normalized attention coefficients (critically, this means that at each training iteration, each node is exposed to a stochastically sampled neighborhood). Furthermore, the GAT architecture employs residual skip connections [16] across the intermediate attentional layers. The model is trained with average dice loss across classes and node input features are standardized, which has proven to achieve better validation performance in this case.

---

[2]We used the code of `https://github.com/tkipf/gcn` and adapted it to handle variable number of meshes. Note that this was possible because all brain meshes have the exact same connectivity pattern.

[3]We used the code of `https://github.com/PetarV-/GAT` and incorporated additional losses.

Table 1: Cross-validation results and comparison on the Human Connectome Project mesh dataset. Results are reported in terms of per-class and average Jaccard index of the two classes of interest.

| Method | neighbor info | global info | node features | Jacc. 44 [%] | Jacc. 45 [%] | mean Jacc. [%] |
|---|---|---|---|---|---|---|
| NodeAVG | ✗ | ✗ | ✗ | $53.0 \pm 2.7$ | $46.9 \pm 5.0$ | $49.9 \pm 2.7$ |
| NodeMLP | ✗ | ✗ | ✓ | $47.8 \pm 2.5$ | $29.6 \pm 3.9$ | $38.7 \pm 2.8$ |
| MeshMLP | ✗ | ✓ | ✓ | $55.8 \pm 3.7$ | $47.9 \pm 2.8$ | $51.8 \pm 2.6$ |
| Jakobsen et al. [21] | ✓ | ✗ | ✓ | $\mathbf{56.4 \pm 2.9}$ | $\mathbf{48.3 \pm 5.3}$ | $\mathbf{52.4 \pm 2.6}$ |
| GCN | ✓ | ✓ | ✓ | $62.0 \pm 3.0$ | $\mathbf{54.2 \pm 4.0}$ | $\mathbf{58.1 \pm 3.1}$ |
| GAT-const | ✓ | ✓ | ✓ | $60.2 \pm 3.0$ | $51.5 \pm 6.5$ | $55.9 \pm 3.9$ |
| GAT | ✓ | ✓ | ✓ | $\mathbf{62.6 \pm 3.4}$ | $52.1 \pm 6.0$ | $57.7 \pm 2.5$ |
| GCN (degree) | ✓ | ✓ | ✓ | $62.8 \pm 2.4$ | $\mathbf{55.7 \pm 4.7}$ | $59.2 \pm 3.0$ |
| GCN (coords) | ✓ | ✓ | ✓ | $\mathbf{64.2 \pm 2.4}$ | $55.2 \pm 5.0$ | $\mathbf{59.7 \pm 3.5}$ |
| GAT (degree) | ✓ | ✓ | ✓ | $63.2 \pm 3.1$ | $53.6 \pm 4.7$ | $58.4 \pm 2.9$ |
| GAT (coords) | ✓ | ✓ | ✓ | $63.5 \pm 2.7$ | $55.0 \pm 4.9$ | $59.2 \pm 2.9$ |

## 3.3   Results

Table 1 summarizes the results obtained by the different proposed methods. Results are reported in terms of Jaccard index of the two classes of interest (areas 44 and 45 of the brain), as well as their average score. We divided the models according to their characteristics, i.e. their ability to exploit different information. We consider three different kinds of information that the models can use: *neighborhood information*, *global information* and *node features*. First, using *node features* implies exploiting them to make the node predictions. Clearly, this is the simplest information to exploit, and is incorporated in the prediction of all models but NodeAVG, which only leverages the node position. Second, *global information* refers to the access to feature information from all nodes in the mesh to predict the segmentation output. This property is found in the MeshMLP baseline, which considers the features of all nodes in a mesh as input. Third and last, *neighborhood information* implicitly provides relational information among nodes, i.e. information about the neighborhood connectivity of each node. This feature is exploited by convolutional-based approaches such as GCN and GAT, which allow parameter sharing across input locations. Note that by stacking multiple convolutional layers and/or expanding the neighborhood around each node, we can enlarge the receptive field of the network, and eventually gain access to an increasingly large portion of the input graph (we report this as global information in Table 1).

As reported in the table, the first group (first four rows) includes all the baselines and state-of-the-art models, which do not exploit local neighborhood information, global information and node features simultaneously. The second group (rows 5 to 7) comprises the models based on graph convolutions, GCNs and GATs. Note that we also report an ablation test on the GAT model, by fixing the attention given to each node to be constant (GAT-const in the table).

Among baseline and state-of-the-art models, NodeMLP exhibits the lowest performance, with an average Jaccard of $38.7\%$. This behavior is expected, since it only processes the features of one node at a time, making node's predictions independent of each other. By contrast, NodeAVG, which ignores the node features and takes into account the node positions exclusively, yields better results, highlighting the importance of the position cue. We argue this is due to existing partial overlap of the brain areas across different subjects. Moreover, producing a mesh level parcellation instead of classifying each node independently also offers a performance boost w.r.t. NodeMLP: see how MeshMLP increases the mean Jaccard score by more than $10\%$, emphasizing the benefits of introducing global information when making predictions. However, the best score within this group is achieved by the approach proposed by Jakobsen *et al.* [21], the average of which is $0.6\%$ above MeshMLP and $2.5\%$ above NodeAVG. It is worth noting that this state-of-the-art method injects local neighborhood information in a post-processing clustering step and uses a larger number of functional features to achieve this result, which could also prove beneficial to other models.

Graph convolutional models increase the overall performance further. GCN and GAT perform on par, improving by a margin of at least $5\%$ with respect to the average Jaccard of the baselines, taking advantage of their access to the underlying mesh structure of the data when making predictions. GAT-const uses the same architecture as GAT but with a constant attention mechanism (assigning the same importance to each neighbor). Given the proposed GAT architecture described in Section 3.2,

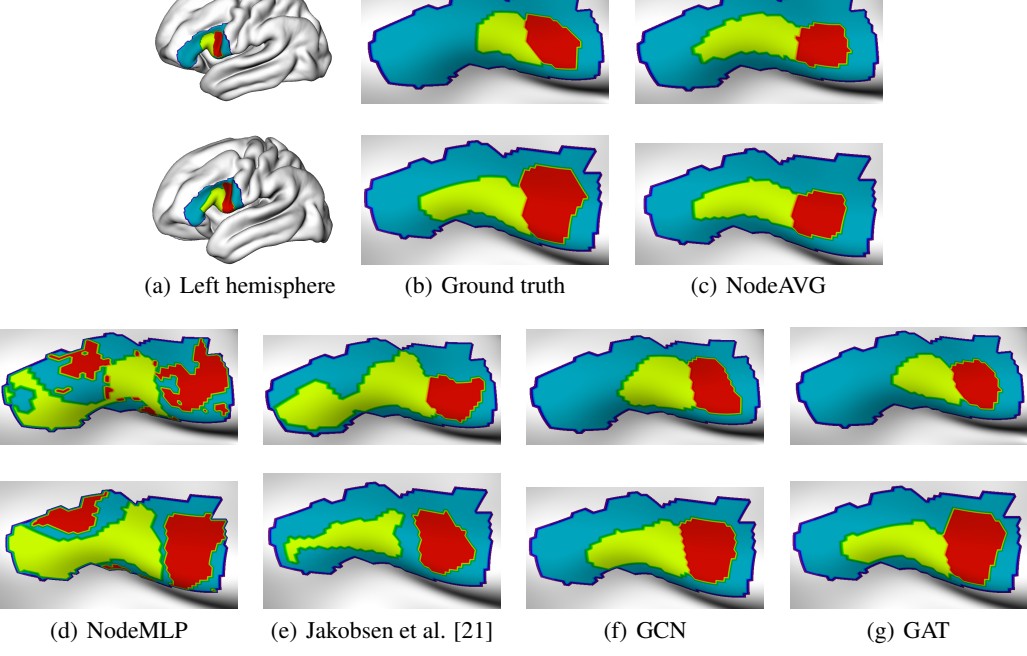

(a) Left hemisphere     (b) Ground truth     (c) NodeAVG

(d) NodeMLP    (e) Jakobsen et al. [21]    (f) GCN    (g) GAT

Figure 1: Area parcellation qualitative results on the test set for two subjects. In all subfigures, top images show data for one subject while the bottom images show the visualization for the second subject. For visualization purposes, we show the segmentation results on a smoothed cortical mesh. Red represents area 44, green represents area 45 and blue represents background.

which considers neighbors within 5-hops of the central node, the GAT-const architecture is limited to applying the same constant attention to all neighbors that are up to 5 steps away. Note that this architecture is less flexible than GCN, which learns one coefficient per polynomial degree.

We also evaluate and compare some of the baselines and state-of-the-art methods to graph convolutional approaches qualitatively. Figure 1 shows some parcellation results produced for two test subjects (area 44 is represented in red, area 45 in green and background in blue). Qualitative results are well aligned with the quantitative ones. As shown in Figure 1(d), NodeMLP produces the noisier parcellation, which could be explained by the lack of neighborhood and global information suffered by this model. Both NodeAVG and Jakobsen et al. [21] do a decent job at segmenting area 44, but largely oversegment area 45 in some cases (see 1(c) and 1(e), respectively). Finally, GCN and GAT show more accurate segmentations for both areas (see 1(f) and 1(g), respectively).

## 4 Discussion

From the experimental results, it is clear that graph-based methods (GCN and GAT) perform better than the baseline approaches. This emphasizes the importance of exploiting the mesh structure in the data, with the baselines either leaving it completely unused, or eliminating all relational information from it.

It is not possible to conclusively determine which graph-based approach has performed better on this dataset, keeping in mind the standard deviations in the performance metrics. However, some conclusions may still be drawn by considering the average performance of each model. The GCN has outperformed the GAT on average by a slight margin (of $0.4\%$), which may be explained by the GCN having direct access to the adjacency matrix and node degree information, while the GAT model only uses the adjacency matrix to mask its self-attention coefficients. This result has implied that, on this task, the benefits of exploiting the regularity in the graph structure may outweigh the value of assigning different importances within the neighborhood. As a result, we have decided to conduct further studies in this direction—namely, ones wherein we have injected the degree information to the features of each node (denoted *degree* on Table 1). As anticipated, this has consistently improved both the GCN and GAT models' predictive power, achieving a rise in average performance

of 1.1% and 0.7%, respectively. Encouraged by this result, we have attempted to introduce additional positional features, i.e. the spatial coordinates of the nodes (denoted *coords* on Table 1). We show that by incorporating these features, we are able to achieve the best per-class and average Jaccard performance compared to previously discussed approaches, with an increase in average performance of 1.6% and 1.5% for GCN and GAT, respectively. Furthermore, it is worth noting that, unlike $k$-step GCN, $k$-step GAT models are not able to give special treatment to different hop neighborhoods.

While these two approaches have been successfully tested on the relatively specific challenge of parcellating cortical areas 44 and 45 using structural and functional MRI data, the advantage of this approach is that it can be readily adapted to many other neuroimaging challenges. Similar architectures could be applied to any data modality on the cortical mesh, including Positron Emission Tomography, Diffusion MRI, cytoarchitectural and even genetic data. Furthermore, while aim of these models was to parcellate the cortex into functionally discrete areas, the methods utilized are general, so they can be applied to many other tasks. With suitable training data, they could be used to segment lesions in neurological diseases such as epilepsy [2] or multiple sclerosis [30].

The networks were more consistently able obtain a higher performance in parcellating area 44 than area 45. This could be for several reasons. First is that the functional features were chosen from a total of 32k rsfMRI internodal correlation scores due for being distinct between areas 44 and 45, not necessarily to differentiate these areas from their surrounding cortex. Thus addition of extra functional features might better isolate area 45 from its other neighboring areas, particularly area 47/12 which has a similar functional connectivity profile [21].

A second limitation of the proposed cortical mesh segmentation approach is that it is currently limited to subsets of the total cortical mesh in the form of patches. A full cortical mesh may have up to 160k nodes, and current graph convolutional approaches are limited by the amount of memory in modern GPUs. Whereas the two evaluated models can scale to larger meshes than the ones used, they may not be able to operate on a 160k nodes mesh. To deal with large meshes, possible solutions that we have not explored in this work consist on using a subsampling operator to reduce the resolution of meshes or operating on smaller patches of the whole mesh.

## 5   Conclusions

In this paper, we tackled the important problem of parcellation of the cerebral cortex into functionally discrete areas, with main focus on the Broca's area. We framed the problem as a mesh segmentation task, and addressed it by taking advantage of recent advances in generalizing convolutions to the graph domain. In particular, we proposed to assess graph convolutional networks and graph attention networks as alternatives to the current state-of-the-art [21], which relies on node features without exploiting the underlying structure of the data to make predictions. We evaluated the proposed models on the HCP dataset and successfully reported state-of-the-art performance, highlighting the importance of both local neighborhood as well as contextual information.

Therefore, we demonstrated the potential of recent advances in generalizing convolutions to the graph domain and their applicability to tackle important neuroimaging challenges, showing that we can improve standard practices in the analysis of cortical meshes and making the models readily adoptable to investigate a myriad of other neuroscientific questions, such as disease diagnosis, lesion characterization and developmental/pathological prognoses.

**Acknowledgments**

The authors would like to thank the developers of TensorFlow [1]. We acknowledge the support of the following agencies for research funding and computing support: CIFAR, Canada Research Chairs, Compute Canada and Calcul Québec, as well as NVIDIA for the generous GPU support. PV has received funding from the European Union's Horizon 2020 research and innovation programme PROPAG-AGEING under grant agreement No 634821. KW received funding from Montreal Neurological Institute and University of Cambridge (RG90792 RRZD/026). GC received funding from the Spanish project TIN2015-65464-R (MINECO/FEDER). Data were provided [in part] by the Human Connectome Project, WU-Minn Consortium (Principal Investigators: David Van Essen and Kamil Ugurbil; 1U54MH091657) funded by the 16 NIH Institutes and Centers that support the NIH Blueprint for Neuroscience Research; and by the McDonnell Center for Systems Neuroscience at Washington University. Finally, the authors would like to thank Joseph Paul Cohen for support.

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
