# OpenReview forum: "Convolutional neural networks for mesh-based parcellation of the cerebral cortex"
_MIDL.amsterdam/2018/Conference — MIDL 2018 Oral_

### Review · AnonReviewer1 · 2018-05-08
**While the application of graph-based convolutional neural network to brain parcellation is novel, the evaluation remains rather limited and shows only limited promise for achieving a high segmentation accuracy for more complex brain parcellation tasks.**

**Rating:** 2
**Confidence:** 2

**Review:**

Summary:
This paper applies existing graph-based convolutional neural network approaches to cerebral cortex parcellation study to compare their segmentation performances. NodeMLP, NodeAVG, MeshMLP, GCN, and GAT were applied to HCP dataset. Segmentation targets were two adjacent cytoarchitectonic areas 44 and 45.

Pros:
-New application of the graph-based convolutional neural network approaches is proposed. These approaches perform segmentation of areas in the cerebral cortex.
-Investigation and comparison of segmentation performances of the graph-based approaches.

Cons:
-No new method is proposed in this paper.
In the experiments, only small regions around areas 44 and 45 were processed. Performing three-classes segmentation task in the small regions seems a relatively easy task. However, the segmentation performance only achieved up to 59.7%. From this results, the evaluated graph-based segmentation approach show only limited promise to be able to segment in more broad areas on the cerebral cortex. Some questions still remain:
-Why the areas 44 and 45 were selected as segmentation target?
-Performing segmentation in small regions around areas 44 and 45 is really meaningful?
-Is it possible to segment areas in more broad areas on the cerebral cortex with reasonable segmentation performance?

Overall opinion:
While the application of graph-based convolutional neural network to brain parcellation is novel, the evaluation remains rather limited and shows only limited promise for achieving a high segmentation accuracy for more complex brain parcellation tasks.


**Special Issue:**

No

---

### Review · AnonReviewer3 · 2018-05-09
**Application of graph based CNN to brain parcellation problem is interesting but lacks novelty in terms of the methods applied. The evaluation is satisfactory but the presented work lacks substantiation for the experimental methods used.**

**Rating:** 3
**Confidence:** 2

**Review:**

The paper addresses the problem of brain parcellation by applying existing graph based CNN techniques -GCN and GAT to parcellate areas 44 and 45 of the brain. The application of the graph based CNN methods is theoretically straight forward and hence the work lacks novelty. The results obtained by the proposed method show increase in Jaccard index compared to state-of-the-art non deep learning methods.

pros
-Application of graph based CNN on cerebral cortex parcellation
-Satisfactory evaluation of graph based CNN methods to non deep learning based methods in literature
-Good investigative analysis of various iterations of GCN and GAT methods for superior quantitative results

cons
-The paper does a poor job of explaining the experiments
-No network architecture is provided
-" For all models, we have experimented with different configurations, selecting the best architecture and optimization hyper-parameters in terms of validation Jaccard index.", "The optimized loss function is either the dice loss [7], extended to multiple classes by averaging class-specific dice losses, or a crossentropy loss, according to the best validation results."=> This is very adhoc, it is understandable it is not feasible to provide all the information, but the authors can do a better job in giving a high level idea on which different architecture/loss functions were used for what method. Was this determined by conducting only a 1 fold experiment or the determination is also based on a 10-fold cross validation? What are the different models investigated for non deep learning graph based approaches since they are based on MLP.
-For GAT training what was the reason for choosing neighbors withing 5-hops? why not 4 or 6? Does it effect performance? if so how?
-"A second limitation of the proposed cortical mesh segmentation approach is that it is currently limited to subsets of the total cortical mesh in the form of patches"=> when are the patches extracted, how are they extracted?
-All the subjects had the same mesh structure. How will the algorithm perform if that is not true?
-How significant is the improvement in accuracy from graph based CNN methods?


**Special Issue:**

No

---

### Review · AnonReviewer2 · 2018-05-09
**Graph-based CNNs work well for segmenting (a small part of) the cerebral cortex.**

**Rating:** 4
**Confidence:** 2

**Review:**

The authors provide a detailed and thorough investigation of graph-convolution methods for calculating a mesh of the cortical surface.

+ the comparison of various methods is thorough and well-done.
+ the paper reads well (although the results are not reproducible from the text, and there is no promise of making the code open)
+ these preliminary results are promising for the chosen task.

- the justification for only using one small area of the cortex is slightly lacking. While Broca's area may be interesting, it seems that the approach could have been tested on other areas as well -- even choosing another two bordering regions in the dataset. This would have bolstered my confidence in the method's generaliseability.

- while analysis on the cortical surface is indeed common, the popularity of FreeSurfer is partly in its applicability to a wide variety of data, from numerous scanners/protocols. This experiment is only carried out on the highest quality data (Human Connectome Project). How would it fare under other conditions? Unfortunately, there is no attempt to use the method on another dataset (e.g. the IXI dataset: https://www.nitrc.org/projects/ixi_dataset/ ).

- there could have been more presentation of experimental details (e.g. the architecture used), or at least an indication that the code will be released publicly.


**Special Issue:**

Yes

---

### Comment · ~Guillem_Cucurull1 · 2018-05-14
**Answer to the reviewers**

First of all, we would like to thank the reviewers for their time and the useful comments they provided. As there is no rebuttal phase, we will address the main concerns shared by reviewers in this answer.

Regarding the architectural details and the code availability, the code of both GCN and GAT is publicly available. We will share the training scripts with the architectural details and loss functions upon publication. Hopefully, this will ease the understanding of the proposed architectures. Also, with respect to the neighborhood size chosen for GAT, it is just another hyperparameter, like GCN’s degree, that is chosen by experimenting with different configurations.

As for the mesh structures, if they were different for each subject, GAT would still be applicable, since it does not rely on knowing the mesh connectivity upfront. This is not necessarily the case for other methods.

Regarding the experiments section, we have utilized different architectures, such as NodeMLP, MeshMLP, NodeAVG, GCN and GAT, which have different properties, to showcase the potential of models directly exploiting the underlying mesh structure of the data. The paper provides an in depth analysis of graph convolutional based models to tackle the problem of cortical mesh parcellation. While there are previous attempts using MLP-like architectures to address this problem, to the best of our knowledge the application of recent graph convolutional models to this domain is novel and opens the door to many research directions on how to enhance those models to improve their performance on mesh parcellation tasks.

We have experimented with segmenting areas 44 and 45 from Broca’s region, which is interesting because the same region has a high degree of variability between subjects. As it can be seen from previous approaches Jakobsen et al. [21], even though that the region is segmented into 3 different areas, the task is not trivial, and we improve the previous results by more than 7% points. Also, more regions are being manually segmented at the moment, and the proposed method can seamlessly be applied to different regions in the future. Therefore, we believe our work is important since it proves that graph-based deep learning methods can be applied to improve cortical mesh parcellation results, and encourages more work in this direction.

---

### Comment · ~Bram_van_Ginneken1 · 2018-05-18
**Selection for longlist for special issue Medical Image Analysis**

Dear authors,

Congratulations on your acceptance to MIDL! We have selected your paper on the longlist for the Medical Image Analysis Special Issue. Please read this page:
https://midl.amsterdam/special-issue-in-medical-image-analysis/
Please answer the three questions that are listed on that page about your interest in submitting to the special issue, potential overlap with other publications, and related publications.

You can post your answer here directly below on openreview.net, or mail me directly at bram.vanginneken@radboudumc.nl.

Best regards, Bram

---

### Decision · Program_Chairs · 2018-05-15
**Paper55 Acceptance Decision**

Oral